# Calcifediol for Use in Treatment of Respiratory Disease

**DOI:** 10.3390/nu14122447

**Published:** 2022-06-13

**Authors:** Marta Entrenas-Castillo, Lourdes Salinero-González, Luis M. Entrenas-Costa, Rubén Andújar-Espinosa

**Affiliations:** 1Pneumology Department, Hospital QuironSalud, 14004 Cordoba, Spain; marenca@gmail.com (M.E.-C.); luis.m.entrenas@gmail.com (L.M.E.-C.); 2School of Medicine, University of Córdoba, 14071 Cordoba, Spain; 3Endocrinology and Nutrition Department, Hospital Universitario Reina Sofía, 30003 Murcia, Spain; lourdessalinero@hotmail.com; 4Pneumology Department, Hospital Universitario Reina Sofía, 14004 Cordoba, Spain; 5Pneumology Department, Hospital Clínico Universitario Virgen de la Arrixaca, 30120 Murcia, Spain; 6Medicine Department, University of Murcia, 30120 Murcia, Spain

**Keywords:** calcifediol or 25-hydroxyvitamin D3, calcitriol or 1α, 25-dihydroxyvitamin D3, vitamin D endocrine system, vitamin D receptor, VDR, vitamin D3, cholecalciferol, respiratory diseases, bronchial asthma, COPD, COVID-19, SARS-CoV-2

## Abstract

Calcifediol is the prohormone of the vitamin D endocrine system (VDES). It requires hydroxylation to move to 1,25(OH)2D3 or calcitriol, the active form that exerts its functions by activating the vitamin D receptor (VDR) that is expressed in many organs, including the lungs. Due to its rapid oral absorption and because it does not require first hepatic hydroxylation, it is a good option to replace the prevalent deficiency of vitamin D (25 hydroxyvitamin D; 25OHD), to which patients with respiratory pathologies are no strangers. Correcting 25OHD deficiency can decrease the risk of upper respiratory infections and thus improve asthma and COPD control. The same happens with other respiratory pathologies and, in particular, COVID-19. Calcifediol may be a good option for raising 25OHD serum levels quickly because the profile of inflammatory cytokines exhibited by patients with inflammatory respiratory diseases, such as asthma, COPD or COVID-19, can increase the degradation of the active metabolites of the VDES. The aim of this narrative revision is to report the current evidence on the role of calcifediol in main respiratory diseases. In conclusion, good 25OHD status may have beneficial effects on the clinical course of respiratory diseases, including COVID-19. This hypothesis should be confirmed in large, randomized trials. Otherwise, a rapid correction of 25(OH)D deficiency can be useful for patients with respiratory disease.

## 1. Introduction

Vitamin D is a threshold nutrient, which is conditionally indispensable, and therefore meets the criteria to be considered a vitamin in situations where endogenous synthesis is profoundly impaired (such as in vulnerable groups of people, including newborns, infants, people who cover their skin for religious or cultural reasons or those with dark skin, especially if they live in northern countries, and in elderly people confined to the home, such as nursing home residents, etc.) [1]. Vitamin D is part of an endocrine system, the vitamin D endocrine system (VDES), in much the same way as other steroid hormones. Calcifediol is a prohormone of this endocrine system and is produced mainly after hepatic hydroxylation of vitamin D3 (cholecalciferol). In order to convert to the active hormonal form (calcitriol), new hydroxylation by the enzyme 25(OH)D-1α-hydroxylase is necessary [2,3] (Figure 1).

Vitamin D3 is synthetized principally (80–90%) by ultraviolet B (UV-B) radiation (280–315 nm) through a non-enzymatic reactions with 7-dehydrocholesterol in the skin and in minimal amounts from the diet, because only a few foods contain it, mainly as vitamin D3 (or its metabolite calcifediol [1]). Another isoform, which is available in nutritional and pharmacological forms, “vitamin” D2, is found in small amounts in foods of plant origin, yeasts and fungi [4,5].

The vitamin D3 produced in the skin is transported directly through the blood, bound mainly to vitamin D-binding protein (DBP) encoded by the GC gene. Dietary vitamin D is absorbed by intestinal cells and is transported by chylomicrons into the lymph before reaching the bloodstream, where it binds to DBP [6]. The absorption efficiency of cholecalciferol is good but not complete, as the average absorption is less than 80% in healthy individuals [7]. Intestinal absorption of calcifediol is over 93% in healthy individuals and (almost) equally efficient in patients with malabsorption [8].

Cholecalciferol is the biologically inactive nutrient of the VDES and requires metabolic modifications to be converted into the hormonal form of the VDES. Through the action of the hepatic 25-hydroxylase enzyme (CYP2R1/CYP27A1 and others), hydroxylation of cholecalciferol enables the synthesis of 25-hydroxyvitamin D3 (25(OH)D3; calcifediol), a prohormone of the VDES, which has a long half-life (2–3 weeks).

Calcifediol is a substrate in the synthesis of 1,25(OH)2D3, or calcitriol, by 1-α-hydroxylase (CYP27B1), which is used in the kidneys for its endocrine actions, and in cells of multiple tissues, organs, and systems (e.g., skin, parathyroid gland, breast, colon, prostate, lung, immune system, and bone) for its auto/paracrine actions (Figure 1). Calcitriol is a VDES hormone with a short half-life (5–8 h).

Renal 1-α-hydroxylase is regulated through a feedback mechanism by parathyroid hormone (PTH). At higher levels, it increases calcitriol production, which, in turn, inhibits PTH production. Hypophosphatemia and fibroblast growth factor 23 (FGF23) also regulate 1-α-hydroxylase, increasing and decreasing, respectively, the production of calcitriol [6].

Figure 2 shows the main routes of oxidation of vitamin D3. Cytochrome P450 (CYP450) and enzymes that catalyze the different steps (CPY2R1, CYP34A, and CYP27A1) are required throughout the process and are expressed both in the liver and in the respiratory tract [9]. Calcifediol can undergo alternative hydroxylation to make inactive compounds through the catalytic action of various enzymes (CYP241A, CYP3A4, and CYP3A5) expressed not only in the liver, but also in the airways [10].

Nevertheless, there is no clear consensus on the optimal serum level of 25(OH)D. The United States Institute of Medicine (IOM) (subsequently renamed the National Academy of Medicine), in 2011, specified a serum 25(OH)D level of ≥50 nmol/L (20 ng/mL) as the lower target value for an adequate vitamin D supply. Shortly after, the D-A-CH nutrition societies (D-A-CH: Germany, Austria, Switzerland), the Scandinavian nutrition societies, the German Osteology governing body (DVO) and the European Society for Clinic and Economic Aspects of Osteoporosis and Osteoarthritis adopted the IOM guidelines. In contrast, the Endocrine Society and the International Osteoporosis Foundation consider an adequate vitamin D supply to be guaranteed at levels of at least 75 nmol/L (30 ng/mL). Several medical societies and non-governmental organizations such as the American Geriatrics Society (AGS) and the International Osteoporosis Foundation (IOF) have adopted the ES guidelines [11]. Although there is no clear consensus on the optimal serum level of 25(OH)D, a value of ≥30 ng/mL (75 nmol/L) has been considered necessary to ensure optimal health [12]. The prevalence of 25(OH)D serum levels below the recommended levels is high worldwide [13], even in regions such as the Mediterranean basin, where the number of hours of sunshine is high throughout the year [14].

Patients with chronic respiratory diseases such as asthma [15] or COPD [16] are not free from this deficit, and it may even be more prevalent. In blood, the poorly water-soluble metabolites of VDES are 88% transported by vitamin D transport protein (DBP (GC)), with varying degrees of affinity, i.e., higher for calcitriol and lower and decreasing for calcifediol, 24,25 hydroxyvitamin D3, and cholecalciferol. Ten percent is transported by albumin, with only 1–2% circulating in free form [3].

1,25(OH)2D3 binds with high affinity to its receptor (vitamin D receptor (VDR)), whereas calcifediol, 24,25 hydroxyvitamin D3, and 1,24,25 trihydroxy vitamin D and other metabolites have much lower affinity [3]. The VDR belongs to the superfamily of nuclear steroid receptors that use the same heterodimer partner (RXR) and co-activators or repressors that regulate transcription in a large number (~3%) of genes. It has a broad spectrum of functional activities [3].

In catabolism, the VDES uses the enzyme 24-α-hydroxylase (CYP24A1), both in the kidneys (by endocrine control) and in other cells and tissues of the system, to form inactive metabolites from 25(OH)D and 1,25(OH)2D, namely, 24,25-dihydroxyvitamin D and 1,24,25-trihydroxyvitamin D, which metabolize to calcitroic acid. In addition, it uses other glucuronic or sulphate metabolites (eliminated mainly by bile), thus constituting an important system of catabolic regulation of VDES metabolism [3].

Classically, the action of VDES has been limited to its effects on calcium homeostasis and bone health. However, our knowledge of the extra-skeletal effects of the system [6] has grown with the discovery of the VDR and the activity of the enzymatic machinery of the system in activation and catabolism in various organs [17]. For this reason, it has been implicated as a factor that decreases risk in numerous conditions, including the most common tumors and autoimmune, infectious, and cardiovascular diseases [5].

Lung cells, and more specifically type II alveolar cells, epithelial cells, smooth muscle cells, alveolar macrophages, NK cells, mast cells, dendritic cells, and lymphocytes, constitutively or after activation, where appropriate, express VDR and the 1-alpha-hydroxylase, which can function as the origin and target of the vitamin D hormone [18]. Several of its key genes such as surfactant factor and cathelicidin are regulated by 1,25(OH)2D. 1-alpha-hydroxylase activity in lung tissue may explain the actions of calcifediol and its higher efficiency with respect to cholecalciferol in respiratory diseases, as calcifediol restores normal blood levels of 25(OH)D more quickly than cholecalciferol, with a 3- to 5-fold greater potency for raising these levels [19]. Moreover, observational and intervention studies suggest that the 25(OH)D status may have clinical implications for the respiratory system [20]. From this rationale, the aim of this narrative revision is to report the current evidence on the role of calcifediol, the major circulating metabolite of vitamin D, for respiratory diseases.

## 2. Vitamin D Deficiency and Asthma

The benefits of the VDES in the treatment of asthma have been a controversial issue for many years, with multiple studies analyzing different clinical variables and marked variability in methodology, thus making it difficult to perform quality meta-analyses. 

In asthma, the VDES acts via several mechanisms. For example, it can act on innate immunity, modulating the production of proinflammatory cytokines and increasing the production of antimicrobial peptides. It can also impact adaptive immunity through a direct effect on T cells, thus influencing the Th2 response, reducing IgE production, and increasing IL-10 synthesis, as well as decreasing IL-17 levels in patients with severe asthma [21]. The VDES may also enhance the actions of regulatory T cells (Treg), improve corticosteroid response by restoring the induction of IL-10 in asthmatics, and decrease airway smooth muscle mass. These effects suggest that vitamin D could be beneficial for asthmatics.

The mechanisms of action that could explain this association between vitamin D levels and asthma were recently reviewed by Pfeffer and Hawrylowicz [22], who concluded that “vitamin D beneficially modulated diverse immunological pathways in heterogeneous asthma endotypes, regulating the actions of lymphocytes, mast cells, antigen-presenting cells and structural cells to dampen excessive inflammatory responses” and that “similar mechanisms, and effects on fetal lung development, likely underlie a primary prevention therapeutic role in pregnancy for vitamin D to reduce the development of asthma in children”.

In aeroallergen-triggered asthma, there is a failure of immune responses regulated by diverse classes of regulatory Treg and suppression of inappropriate anti-gen-dependent responses to aeroallergens. Adaptive immunity and vitamin D have proven to play vital roles in Treg responses [23]. Moreover, vitamin D can regulate IgE production [22].

In asthma, Naive T lymphocytes are inadequately prepared to respond to inhaled aeroallergens with Th2 polarity, and upon restimulation by inhaled aeroallergens, they produce Th2 cytokines, IL-4, IL-5 and IL-13, that promote the pathological mechanisms of asthma. asthma. IL-4 promotes IgE production by B lymphocytes and coats mast cells, leading to a rapid release of more proinflammatory mediators in asthma. Adaptive immune responses are regulated by distinct classes of regulatory T cells (Tregs) that express Foxp3 and IL-10-secreting Tregs, and their coordinated action in healthy individuals ensures tolerance to harmless antigens [22].

Vitamin D has a vital role in Treg responses. Urry et al. [24] demonstrated in vivo that vitamin D is positively correlated with the frequency of Foxp3+ Tregs and IL-10 levels in patients with asthma, which would improve tolerance to allergens in these patients as well as the response to corticosteroids. Vitamin D suppresses IgE production by human B lymphocytes in vitro and increases IL-10 production, promoting a regulatory phenotype of B lymphocytes, which could decrease the inflammatory response in asthmatic patients [25].

In nonallergic asthma, vitamin D has been shown to modulate the epithelial response to stimulation with a potentially anti-inflammatory role [26].

There are multiple observational studies on vitamin D and asthma in children. No relationship has been found between serum 25(OH)D deficiency in maternal plasma, the umbilical cord, or childhood and the subsequent incidence of asthma in children [27,28,29,30]. However, in 2012, Bener et al. [31] carried out a study with 969 asthmatic children and their healthy controls, finding that 25(OH)D deficiency was more frequent in asthmatic children (AOR = 4.82; 2.41–8.63). Findings for the relationship between asthma severity and serum 25(OH)D deficiency are contradictory. An inverse relationship has been found between serum 25(OH)D levels and the prevalence of severe asthma (OR 0.49; 95% CI 0.25–0.95) in a subgroup of 4-year-old children, while in a subgroup of 8 year olds, opposite, although not significant, findings were reported [32]. Asthma attacks were associated with serum 25(OH)D deficiency by Brehm et al. [33] in 2009 (OR 0.38; 95% CI: 0.20–0.67) and 2012 [34] (OR 0.05; 95% CI: 0.00–0.71) and by Beigelman et al. [35] (OR 0.60; 95% CI: 0.39–0.92).

In adults, no clear relationship has been found between 25(OH)D levels and the incidence and prevalence of asthma.

In 2010, Sutherland et al. [36] carried out a cross-sectional study of 54 non-smoking asthmatic patients over the age of 18, in which they measured the relationship between serum levels of 25(OH)D and the values obtained in pulmonary function testing (FEV1 and methacholine test). The authors found a significant association between higher serum levels of 25(OH)D and better results in pulmonary function tests. Thus, for every 1 ng/mL increase in 25(OH)D values, an increase of 22.7 ± 9 mL in FEV1 values was observed (*p* = 0.02).

In 2013, Korn et al. [37] showed that 25(OH)D deficiency was common in asthmatic patients (67%) and reported that plasma 25(OH)D levels were lower with greater severity (OR 1.9; 95% CI 1.2–3.2) and poorer asthma control (OR 2.1; 95% CI 1.3–3.5). Another non-experimental study found an inverse relationship between the proportion of asthmatics with attacks and 25(OH)D serum levels (*p* < 0.001) [38].

All these data support the hypothesis that improving suboptimal vitamin D status could be effective in the prevention and treatment of asthma. While several randomized clinical trials (RCTs) have attempted to demonstrate the benefits of the VDES in people with asthma, the results have been contradictory.

A meta-analysis that included three RCTs performed in children showed a protective effect of vitamin D in reducing asthma attacks (RR 0.41; 95% CI 0.27–0.63; *p* ≤ 0.001) [39]. However, the studies included in this meta-analysis did not exclusively analyze children with 25(OH)D serum levels, thus indicating that these results could have been even better if all the patients included had had 25(OH)D deficiency. The results of an RCT from 2021 that included only children with serum 25(OH)D deficiency showed no improvement in asthma control or in problems of supplementation compared with placebo [40]. None of these studies used calcifediol for vitamin D treatment, although adding calcifediol would show the usefulness of this approach in asthmatic children owing to the presence of the VDR and 1alpha-hydroxylase in the respiratory tract.

Many adult RCTs were based on different methodologies and different primary endpoints. These have also generated contradictory results.

In 2014, the multicenter, double-blind VIDA study [41] included symptomatic asthma patients with 25(OH)D deficiency. The main variables were the rate of therapeutic failure and the rate of exacerbations. No significant differences were found, although treatment with vitamin D enabled a 25% reduction in the dose of inhaled corticosteroid. In this same year, Arshi et al. [42] published the results of an RCT that included asthmatic patients with and without 25(OH)D deficiency. The authors investigated whether administration of vitamin D improved the response to long-term treatment with inhaled corticosteroids in asthmatic patients and required at least 24 weeks of vitamin D to be able to observe these effects.

An RCT published in 2015 included asthmatic patients with serum 25(OH)D deficiency. Its main variable was the decrease in eosinophilia in induced sputum, which was significantly reduced in the group treated with vitamin D compared with the placebo group. In addition, a significant improvement in asthma control measured using the Asthma Control Questionnaire was observed in the supplemented group compared with those who received placebo. That same year, the ViDiAs study [43] selected asthmatic patients with or without 25(OH)D deficiency and attempted to determine whether vitamin D was able to increase the time before the first severe asthma attack. The study did not show any differences with placebo, even though the prevalence of serum 25(OH)D deficiency was high (83%).

Despite these contradictory results, a meta-analysis [44] published in 2019 concluded that treatment with vitamin D could reduce the annual rate of exacerbations and exert a positive effect on pulmonary function in patients with airflow limitation. In patients deficient in 25(OH)D, no significant differences were found in control, determination of FeNO, IL-10, or adverse effects.

All these studies used treatment with vitamin D other than calcifediol. As mentioned above, calcifediol quickly corrects 25(OH)D levels. Consequently, in studies with a short follow-up period, treatment with calcifediol could ensure that associated clinical outcomes were observed in patients with asthma.

The ACVID clinical trial [45], which was published in 2020, included 112 asthmatic patients with 25(OH)D deficiency and was based on a short follow-up period of six months. The authors administered calcifediol to treat 25(OH)D deficiency. The primary endpoint was improvement in asthma control, measured using the Asthma Control Test (ACT). A statistically significant clinical improvement was observed in the intervention group (+3.09) compared with the control group (−0.57) (difference 3.66, 95% CI 0.89–5.43; *p* < 0.001), as measured using ACT scores. Among the secondary endpoints, a significant improvement in quality of life was found in the intervention group (5.34) compared with the control group (4.64) (difference 0.7, 95% CI 0.15–1.25; *p* = 0.01).

A meta-analysis published in 2021 [46] showed that treatment with vitamin D significantly reduced the risk of asthma exacerbation (RR 0.70, 95% CI 0.59–0.83; *p* < 0.05), although no improvement in asthma control or lung function was observed.

Beneficial vitamin D effects by treating asthma have primarily been seen in children. Maretzke et al. concluded in 2020 that adequate serum levels of vitamin D in children could reduce the risk of asthma exacerbations, while in the adult population, the data are insufficient to draw reliable conclusions [1].

The benefits of vitamin D in the treatment of asthma, which have been observed in clinical practice, are now backed by scientific evidence, although new standardized studies are still needed to ensure rapid correction of serum 25(OH)D deficit to demonstrate all the benefits of vitamin D in the prevention and treatment of asthma. In this sense, calcifediol could help demonstrate these clinical benefits.

## 3. Vitamin D Deficiency and Chronic Obstructive Pulmonary Disease

Low 25(OH)D levels could result from the pathogenic mechanisms of chronic obstructive pulmonary disease (COPD), which disturb the metabolism of vitamin D, thus reducing synthesis or increasing catabolism. Furthermore, the lack of activity typical of COPD could reduce exposure to sunlight. It remains to be clarified whether the disease alone is the cause of low vitamin D levels, or whether these are an essential contributing factor in the development of respiratory disease [47].

Despite the vast available bibliography for VDES and COPD, the results are drastically reduced if we focus on calcifediol. In any case, given that studies evaluate the increase in 25(OH)D levels, it is irrelevant whether levels are reached with cholecalciferol or calcifediol, since cholecalciferol must be converted to calcifediol. Thus, the results observed for cholecalciferol could be extrapolated to calcifediol; however, this should be confirmed in future clinical trials. Furthermore, available papers are not homogeneous in terms of patient inclusion criteria and differ in terms of the selected primary variable (pulmonary function (FEV1) or exacerbations) and duration (generally 6 to 12 months) [48].

A meta-analysis by Zhu et al. [49] including 4818 COPD patients and 7175 controls from 21 studies concluded that 25(OH)D levels were inversely correlated with the risk of COPD, its severity, and the development of exacerbations.

Lokesh et al. [50] recently reported their results for serum 25(OH)D levels among 100 patients with COPD in a rural population in India. The authors reported that 64.5% of patients had levels below 20 ng/mL, despite adequate exposure to sunlight. This percentage was statistically significant when compared to a control population of 100 subjects (OR 5.05, 95% CI 1.4–17.8) [51]. COPD patients with low levels of vitamin D had a threefold increased risk of exacerbation in the previous year, compared to COPD patients without low levels of vitamin D (OR 3.51, 95% CI 1.27–9.67). Serum 25(OH)D levels below 28.81 ng/mL had the best combined sensitivity and specificity for predicting COPD, while levels below 18.45 ng/mL had the best combined sensitivity and specificity for predicting the risk of acute exacerbation.

These results are similar to those previously published by Malinovschi et al. [52] for a series of 97 patients. The authors found that 96% had low levels of 25(OH)D, which was severe in 36%, with a strong relationship between low levels of 25(OH)D and frequent exacerbations (OR 18.1, 95% CI 4.98–65.8; *p* < 0.001) and hospitalization due to exacerbation (OR 4.57, 95% CI 1.83–11.4; *p* = 0.001).

A potentially similar benefit could be predicted in COPD if the results obtained after treating asthma patients with vitamin D are analyzed in terms of antiviral, antimicrobial, and anti-inflammatory effects [53]; assuming that the risk of respiratory infections [54] and asthma attacks is reduced [55], a potential similar benefit could be predicted in COPD. A meta-analysis analyzing prescription of vitamin D to COPD patients to check whether exacerbations are prevented concluded that, overall, treatment with vitamin D did not lower the rate of moderate or severe exacerbations, although this effect was observed when different subgroups were analyzed, specifically, patients with serum calcifediol levels below 25 nmol/L [56].

The underlying mechanism of action for preventing exacerbations is thought to be a depressed immune response associated with low levels of vitamin D, which favors viral and bacterial infections, known triggers of COPD exacerbations. Replacing levels would reduce exacerbations, thus improving the immune response by upregulating the development of antimicrobial peptides [57].

The Global Initiative for Chronic Obstructive Lung Disease (GOLD) [58] includes the recommendation to assess 25(OH)D serum levels in patients admitted to hospital for an exacerbation systematically, advising prescription of vitamin D if necessary.

Mathyssen et al. [48] reviewed eight studies carried out with cholecalciferol at variable doses and intervention times. While generally showing a reduction in exacerbations, the results were not unanimous.

This heterogeneous response to treatment with vitamin D is explained by Jolliffe et al. [59], who found an attenuated response to treatment compared with healthy controls, both in asthma and in COPD. The authors were unable to explain the phenomenon only by genetic variations in the 25(OH)D levels. The explanation could be that the relationship between airway inflammation and vitamin D levels is bidirectional. The cytokines ex-pressed in asthma and COPD are mainly TNF, IL-1b, and TGF-b [60], which induce the expression of CYP24A1 and CYP27B1 in vitro [10] (Figure 2). Therefore, Jolliffe et al. [59] hypothesized that low 25(OH)D levels in inflammatory airway diseases (both asthma and COPD) would arise because of an increase in 24-hydroxylation and the 1-alpha-hydroxylation of 25OH (Figure 2). This possibility does not exclude the favorable effects of treatment with vitamin D reducing the risk of exacerbations, as has been shown in interventional studies.

## 4. Calcifediol and COVID-19

VDR and the enzyme CYP27B1 (which hydroxylates calcifediol to obtain calcitriol, the active hormone of the VDES) are expressed in numerous cells, including cells with a well-known role in immunity in the lungs [61], such as alveolar macrophages, dendritic cells, lymphocytes, airway epithelium, alveoli, and endothelial cells, where the VDES regulates the cytokines and metabolic signaling pathways involved in adaptive immunity [62]. The pulmonary epithelium and the immune and vascular systems play a critical role in the development of COVID-19 [63]. Consequently, since the beginning of the pandemic, the effects that the stimuli of VDR would have on the respiratory distress induced by SARS-CoV-2 have been described [64,65], suggesting a possible link between low levels of 25(OH)D and severity of COVID-19, as supported by the more than 1000 references that appear in PubMed with the keywords “COVID-19” and “vitamin D”.

### 4.1. Potential Mechanisms Linking VDES and COVID-19 Infection

Most cells of the immune system express VDR at some point, suggesting that the VDES participates in both innate and adaptive immunity [66].

In monocytes and macrophages, calcitriol induces production of cathelicidin antimicrobial peptide (CAMP) [67], indicating increased intracellular clearance of bacteria and improvement in antiviral defense mechanisms [68]. Cathelicidin not only prevents bacterial and viral proliferation, but also enhances local inflammation and leukocyte migration to improve pathogen clearance [69]. Neutrophils are the main source of cathelicidin, although the precise role of the VDES with respect to neutrophils needs to be elucidated, even though the VDES is thought to decrease the neutrophil inflammatory responses while T lymphocytes are destroying pathogens [70] (Figure 3a,b).

Activated T lymphocytes also express CYP27B1, which means that they have the capacity to produce calcitriol, irrespective of whether they have an adequate supply of calcifediol as a substrate for their synthesis, and, therefore, intracrine activation of the VDR [71]. The effects of calcitriol on T cells are usually indirect through effects on antigen-presenting cells.

The global effect can be summarized (Figure 4) as a change in adaptive immunity, from Th1, Th9, and Th17 lymphocytes to Th2 cells by suppressing the expression of Th1 cytokines (IL-2, IFN-gamma, TNF-alpha), Th9 (IL-9), and Th17 (IL-17, IL-21), thus inducing the expression of Th2 cytokines (IL-4, IL-5, IL-9, I-13).

The response that leads to the activation of the VDR in the immune system shows a delay of several days after contact with the pathogen. The early response to pathogens is not prevented, but it prevents an excessive reaction that would be detrimental due to an excessive increase in local and systemic inflammation [66].

Finally, inactive B cells do not have vitamin D receptors. However, when they are activated, they express the VDR and CYP27B1 [72]. Activation of the VDR stimulates lymphocyte apoptosis, thus preventing the generation of plasma cells and modulating the production of antibodies [73]. It also improves immunoregulation by regulating IL-10 production by B lymphocytes [74].

In general, the actions of the VDES on the immune system can be considered a combination of stimulation of the innate immune response and, with some delay, reduction of acquired immunity.

### 4.2. VDES and the Lungs

Bronchioalveolar epithelial cells express VDR, CYP27B1, CYP24A1, and cathelicidin [75]. Expression of VDR is pronounced in apical epithelial cells, but not in endothelial cells. Expression of cathelicidin is an additional benefit against pathogens. Maternal 25(OH)D deficiency is linked to a decline in lung function and increased risk of early wheezing in childhood [76].

Given that activation of the VDES decreases the inflammatory response induced by viruses [19], VDES could be beneficial for removing pathogens and suppressing the inflammatory response against invaders. It could reduce the remodeling and fibrosis that induce lung disease, and the anti-proliferative and anti-inflammatory effects of VDR are well known [65].

Multiple studies link 25(OH)D status and upper respiratory tract infections. Observational studies found an increased risk of infection in individuals with low serum 25(OH)D levels. At least two meta-analyses show a significant reduction in the risk of upper respiratory tract infection and daily or weekly treatment with vitamin D, especially when the baseline level before entering the study is low [54,77]. In 2017, Martineau et al. [54] carried out a systematic review and meta-analysis of individual participant data to analyze the use of vitamin D to prevent acute respiratory tract infections. The results showed that treatment with vitamin D reduces the risk of experiencing at least one acute respiratory tract infection. In this sense, the authors proposed a new major indication for treatment with vitamin D, namely, the prevention of acute respiratory tract infection. It was shown that the beneficial effect of treatment with vitamin D was stronger in deficient individuals. Another recent systematic review and meta-analysis by Jolliffe et al. [77] further confirmed these results, concluding that treatment with vitamin D is safe and reduced the risk of acute respiratory infections, despite evidence of significant heterogeneity across trials. Regarding symptoms of upper respiratory tract infections, a post hoc analysis performed by Shimizu et al. [78] found that treatment with calcifediol could reduce physical symptoms at the onset of infection, thus improving quality of life.

### 4.3. COVID-19 and VDES

Since the beginning of the pandemic, different mechanisms have been identified through the stimulation of VDR [64,65,79,80,81] by calcitriol. The improvement in innate immunity means that these can improve the body’s response to SARS-CoV-2, consistent with the reduction in upper respiratory tract infections after treatment with vitamin D, especially in individuals with low levels. However, this remains a hypothesis because of the lack of in vitro data on the effect of calcitriol in COVID-19 infection

Both calcifediol and calcitriol have a stimulating effect on cathelicidin, defensins and autophagy, thus improving the elimination of infected cells and their pathogens.

Severe forms of COVID-19, with mainly pulmonary manifestations, but also multisystemic involvement, point to deregulation of the immune response that triggers phenomena such as thrombosis. However, most of the available publications have focused on the effect on the association with hospitalization, admission to the intensive care unit (ICU), and mortality.

The underlying mechanisms of acute respiratory distress syndrome (ARDS) are not exclusive to SARS-CoV-2. The mechanisms implicated are as follows [64]: generation of cytokine and chemokine storms, activation of the renin–angiotensin system, generation of tissue damage by neutrophils involved in the different defense mechanisms, alterations in coagulation mechanisms, and, finally, generation of pulmonary fibrosis.

Activation of the VDR can counteract these mechanisms, as follows:It can reduce the increased inflammation characteristic of ARDS [82] in the same way as corticosteroids. However, unlike these, it does not inactivate innate immunity, but rather helps to stimulate it [83].Calcitriol is a potent agent that reduces levels of the renin–angiotensin system, which are increased in COVID-19 infection. It can also generate tissue damage through the angiotensin receptor. If it is not well regulated by calcitriol, the action of renin and the renin–angiotensin axis is inhibited [84].Neutrophils are a potent antiviral agent, although they can generate tissue damage by releasing cytokines and chemokines in response to infection. Calcitriol modulates the response of neutrophils, thus helping to reduce tissue damage [65].Calcitriol plays a fundamental role as an antithrombotic by reducing the risk of hypercoagulability and pulmonary or systemic thrombosis. Observational data reveal an association between low serum levels of 25(OH)D and the development of thrombotic events in patients with ischemic stroke [85].Finally, calcitriol reduces levels the expression of fibronectin and collagen, thus inhibiting the trans-differentiation of epithelial cells into myofibroblasts [86].

### 4.4. COVID-19 Clinical Data and VDES

Most of the published studies evaluate the risk of COVID-19 infection, its course or outcome, and 25(OH)D. Regarding the results, some publications find a benefit [87,88,89,90,91,92,93,94,95,96,97,98,99,100,101,102,103,104,105,106,107,108,109,110,111,112,113,114,115,116,117,118,119], while others do not [120,121,122,123,124,125,126,127,128,129,130,131,132], there being clear reasons such as the heterogeneity of the patients analyzed, disease severity, the concept of severity used by each author at initiation of the study, the preparation administered (as well as its dose and administration schedule), and the objective of the study (e.g., admission, survival, death, need for admission to the ICU). In addition, most of the designs are observational and generally do not consider comorbidities or concomitant treatment, such as corticosteroids, which could balance the outcome in favor or against one of the study arms [117].

The first published study [102] was observational, based on an increased susceptibility to respiratory infections in patients with low 25(OH)D levels [54]. The hypothesis was that stimulation of the VDR with calcifediol could reduce the potentially deleterious effects of respiratory distress caused by COVID-19, thus avoiding the need for admission to the ICU and mortality [64].

Two reviews analyzed the results of the published studies, although their findings are divergent. Bassatne et al. [133] analyzed the results of 20 studies (13 cross-sectional and 7 cohort studies) and concluded that serum 25(OH)D levels were below 20 ng/mL and that the outcome of COVID infection was uncertain, although there was a higher probability of mortality (OR 2.1, 95% CI 0.9–4.8) for patients with the lowest 25(OH)D serum levels. A non-significant trend was also observed in terms of need for admission to the ICU, mechanical ventilation, and length of hospital stay. In this analysis, the heterogeneity of the studies included the differences in the definitions of vitamin D status, and the objectives set must be considered. In contrast, Wang et al. [134] restricted their meta-analysis to 17 publications in which vitamin D deficiency was significantly associated with higher mortality (OR 2.47, 95% CI 1.50–4.05), a higher hospitalization rate, and a longer length of stay than in patients who had lower levels of normal vitamin D. There were no significant differences with respect to the need for admission to the ICU.

Finally, a meta-analysis by Borsche et al. [135] assessed the association between the COVID-19 mortality rate and 25(OH)D serum levels before admission or on the day of admission, with the regression analysis suggesting that the theoretical point of zero mortality was approximately 50 ng/mL.

Three recent studies show that treatment with calcifediol could reduce the risk of morbidity and mortality in patients with COVID [102,103,136].

The first was a pilot observational study of 76 patients who were randomized 2:1 in the first wave of the pandemic [102]. The objective was to demonstrate the effect of calcifediol on admission to the ICU and mortality rates. In the group that took calcifediol, 2% had to be admitted to the ICU compared with 50% in the control group, which was where the only two deaths in the entire series occurred. Although the group was small, the enormous statistical significance (*p* = 7.7 × 10^−^^7^) enables us to infer that, despite a small imbalance in the risk factors of both groups, treatment with calcifediol was safe [111].

The second study addressed the reduction in mortality. A retrospective study of patients hospitalized for COVID-19 infection in five hospitals in southern Spain [103] included the option of receiving calcifediol in one center, although the treatment was not available in the remaining centers. Other treatments were considered comparable. The odds ratio of death for patients who received calcifediol (5% mortality rate) was 0.22 (95% CI 0.08–0.61) compared with patients who did not receive it (20% mortality rate; *p* < 0.01). In another observational study of 838 patients at Hospital del Mar in Barcelona (Spain) [136], 447 received calcifediol and 20 required admission to the ICU (4.5%), while of the 391 who did not receive treatment, 82 (21%, *p* < 0.001) required admission to the ICU. When the need for ICU admission was adjusted for age, sex, baseline 25(OH)D levels, and comorbidities, treated patients were shown to have a reduced risk of admission to the ICU (OR 0.13, 95% CI 0.07–0.23). The overall mortality was 10%. In the intention-to-treat analysis, 21 of 447 patients (4.7%) treated with calcifediol at admission died compared with 62 of 391 untreated patients (15.9%) (*p* = 0.0001). The results show a reduced risk of mortality with an OR 0.21 (95% CI 0.10–0.43) if calcifediol was administered during admission to hospital; therefore, the need for admission to hospital was significantly reduced.

These results contrast with those described in the multicenter, double-blind, randomized, placebo-controlled clinical trial by Muray et al. [127] that included 240 hospitalized patients with COVID-19 who had moderate to severe COVID-19 expression (the majority). Patients were randomized to receive a single oral dose of 200,000 IU vitamin D3 (*n* = 120) or placebo (*n* = 120), finding no difference in the risk of death, need for assisted ventilation or even the number of days of hospitalization between the two groups. Treatment with calcifediol has an advantage over vitamin D3 supplementation, as it increases serum 25(OH)D concentrations more rapidly, in a matter of hours rather than days (see below). This property is important for treating COVID-19 because the two main effects of vitamin D regarding COVID-19 are reduced viability and replication of SARS-CoV-2 and reduced risk of a cytokine storm [137].

Also studied was the protective role with respect to disease severity. Data from a cohort of 15,968 patients, including all patients hospitalized with a confirmed diagnosis of COVID-19 between January and November 2020 in Andalusia (Spain) [116], revealed 570 patients who were prescribed cholecalciferol and 374 prescribed calcifediol 15 days before hospitalization. When the period was extended to 30 days before hospitalization, the figures increased to 802 and 439, respectively. When treatment was prescribed 15 days before admission, the association between prescription and survival was stronger for calcifediol (HR 0.67, 95% CI 0.50–0.91) than for cholecalciferol (HR 0.75, 95% CI 0.61–0.91). At 30 days, protection diminished, with the results suggesting that the closer the treatment is to hospitalization, the stronger the protective effect. In the Barcelona area, the risk of COVID-19 infection was assessed in patients who had been prescribed vitamin D or calcifediol in the 4 months prior to COVID-19 infection [138]. Cholecalciferol supplementation was associated with slight protection from SARS-CoV2 infection (HR 0.95, CI 95% 0.91–0.98, *p* = 0.004). Patients on cholecalciferol treatment achieving 25OHD levels ≥30 ng/mL had lower risk of SARS-CoV2 infection, lower risk of severe COVID-19 and lower COVID-19 mortality than non-supplemented 25(OH)D-deficient patients (HR 0.66, CI 95% 0.46–0.93, *p* = 0.018). Calcifediol use was not associated with reduced risk of SARS-CoV2 infection or mortality in the whole cohort. However, patients on calcifediol treatment achieving serum 25(OH)D levels ≥30 ng/mL also had lower risk of SARS-CoV2 infection, lower risk of severe COVID-19, and lower COVID-19 mortality compared to 25(OH)D-deficient patients not receiving vitamin D supplements (HR 0.56, CI 95% 0.42–0.76, *p* < 0.001).

In summary, the current evidence supports links between the VDES and COVID-19 and the benefits of treatment with calcifediol to control or treat this condition. Most authors report better prognosis and COVID-19 outcomes with sufficient 25(OH)D serum levels, with or without treatment. Some report no significant differences based on 25(OH)D levels and/or no improvement after treatment. Some even report a decreased incidence of infection because of prior treatment. Future research should focus on establishing the mechanism(s) underlying this association, as well as optimizing treatment doses for maximum benefit to patients once infected. In the meantime, 25(OH)D deficiency should be corrected whenever possible, as calcifediol is safe, and the potential for toxicity is heavily out-weighed by the potential benefits in relation to protection against COVID-19 [139].

## 5. Vitamin D and Other Respiratory Diseases

The potential effect of the VDES in the defense against respiratory infections could arise from its favorable effects on the innate and adaptive immune systems [140,141] (Figure 3). The higher rate of respiratory infections during winter, coinciding with reduced exposure to sunlight and lower serum levels of 25(OH)D, points to a possible relationship between this deficiency and higher susceptibility to respiratory infections. Randomized clinical trials and meta-analyses of treatment with vitamin D have shown only a small reduction in the risk of respiratory infection compared to placebo [77]. Nevertheless, calcifediol can improve the symptoms of upper respiratory infections and the quality of life of affected patients [78].

The VDES may play a role in respiratory infection by enhancing the immune response to *Mycobacterium tuberculosis*. The VDES has been shown to induce in vitro production of antimicrobial products, such as cathelicidin, which inhibits the replication of mycobacteria [142]. Furthermore, polymorphisms of the VDR could influence the risk of tuberculosis, as well as the response to treatment [143]. Available data from clinical trials show conflicting results. Some trials found radiological improvement in tuberculosis patients receiving vitamin D versus placebo [144,145], while others found no significant difference in negative sputum cultures in these patients [146,147]. Finally, a meta-analysis showed that vitamin D3 treatment did not affect sputum culture conversion time in all patients, although it accelerated sputum culture conversion in the subgroup of patients with multidrug-resistant pulmonary tuberculosis [148].

Cystic fibrosis patients have lower levels of 25(OH)D than healthy controls [149]. However, only a few clinical trials have been carried out, with small samples, and there is no evidence of clinical benefit or harm in affected patients [150].

The correlation between vitamin D intake and the development of lung cancer is controversial. Some studies find no relationship between serum 25(OH)D levels and the overall survival rate of patients with lung cancer [151,152], while others show that increased expression of the VDR in lung cancer is associated with a higher survival rate due to a lower proliferative state and cell cycle arrest in the G1 phase [153,154]. A recent meta-analysis showed that vitamin D not only improves the long-term survival of lung cancer patients, but also has a beneficial effect on the incidence of lung cancer [155].

Calcifediol may have some advantages over vitamin D, which gives it a certain superiority for use in vitamin D deficiency replacement therapy in asthma, COPD, COVID-19 and other respiratory diseases: (1) calcifediol induces a more rapid increase in circulating 25OHD than oral cholecalciferol; (2) oral calcifediol is more potent than cholecalciferol; (3) oral calcifediol has a higher rate of intestinal absorption, which confers advantages in cases of malabsorption; (4) oral calcifediol has a linear dose–response curve, independent of initial serum 25OHD; and (5) intermittent administration of calcifediol produces a fairly stable serum 25OHD, compared to fluctuations following intermittent oral cholecalciferol [156]. Therefore, oral calcifediol is more potent than vitamin D3 according to the results of nine RCTs when comparing physiological doses of oral cholecalciferol with oral calcifediol [157] and in clinical trials where the two drugs have been compared head-to-head [158].

In the absorption process, vitamin D3 is incorporated into chylomicrons and enters the lymphatic system. The chylomicrons enter the bloodstream via the superior vena cava. Most of the vitamin D3 is incorporated into body fat. Vitamin D3 that is slowly released from body fat into the circulation is converted in the liver to calcifediol (25OHD3). This is the likely explanation for why vitamin D3 administration takes time to reach a steady-state concentration of 25OHD3. However, calcifediol is more hydrophilic, and therefore, after ingestion, it is absorbed into the venous portal system, immediately increasing the circulating concentration of 25(OH)D [156], which is available within hours as a substrate for calcitriol synthesis in broncho-alveolar lung cells, immune cells or other potential target tissues. This ease of absorption and availability [159] is of great importance in patients with severe malabsorption of any etiology [156].

Furthermore, calcifediol does not require hepatic 25-hydroxylation. Functional impairment of CYP2R1 activity has been reported in patients with obesity, malabsorption [160] or inflammatory lung diseases such as COPD or asthma [45,59].

In summary, calcifediol could play an important role in the treatment of vitamin D deficiency in patients with respiratory diseases. Depending on the availability of clinical trials or not, we can recommend its use with a greater or lesser degree of evidence: clearly in asthma and with somewhat less evidence in COPD. Maintaining adequate 25(OH)D levels may contribute to better control of both pathologies, avoiding exacerbations, and especially avoiding exacerbations of infectious origin. In COVID-19, observational data consistently suggest that calcifediol supplementation may decrease the severity of this disease, as evidenced by a reduced need for intensive care and a decreased risk of mortality. However, we will have to wait for the results of large, randomized trials clinical trials for this recommendation to have a high level of evidence. In the meantime, the practical recommendation should be to systematically determine 25(OH)D serum levels in all patients with chronic respiratory pathologies, and in those with deficient levels, correction of the 25OHD deficit should be recommended by substitution treatment.

## Figures and Tables

**Figure 1 nutrients-14-02447-f001:**
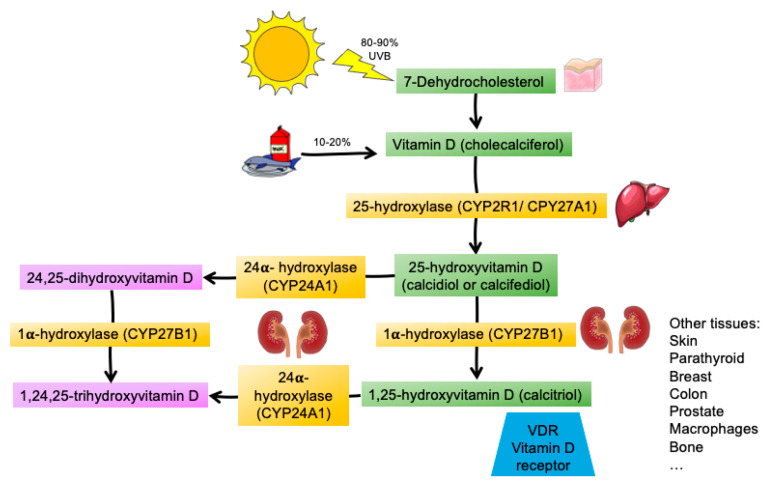
The vitamin D endocrine system metabolism.

**Figure 2 nutrients-14-02447-f002:**
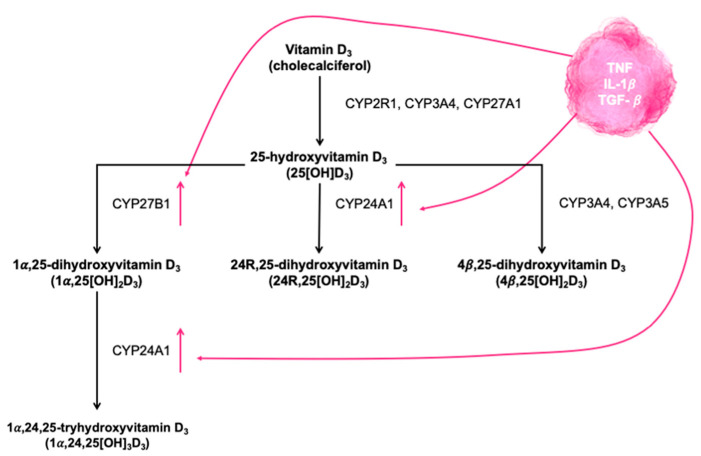
The main oxidation pathways of vitamin D3 and the enzymes catalyzing each step. The cytokine profile that dominates inflammatory respiratory diseases (asthma and COPD) exerts different actions on oxidation enzymes, enhancing the passage to inactive compounds without biological effects.

**Figure 3 nutrients-14-02447-f003:**
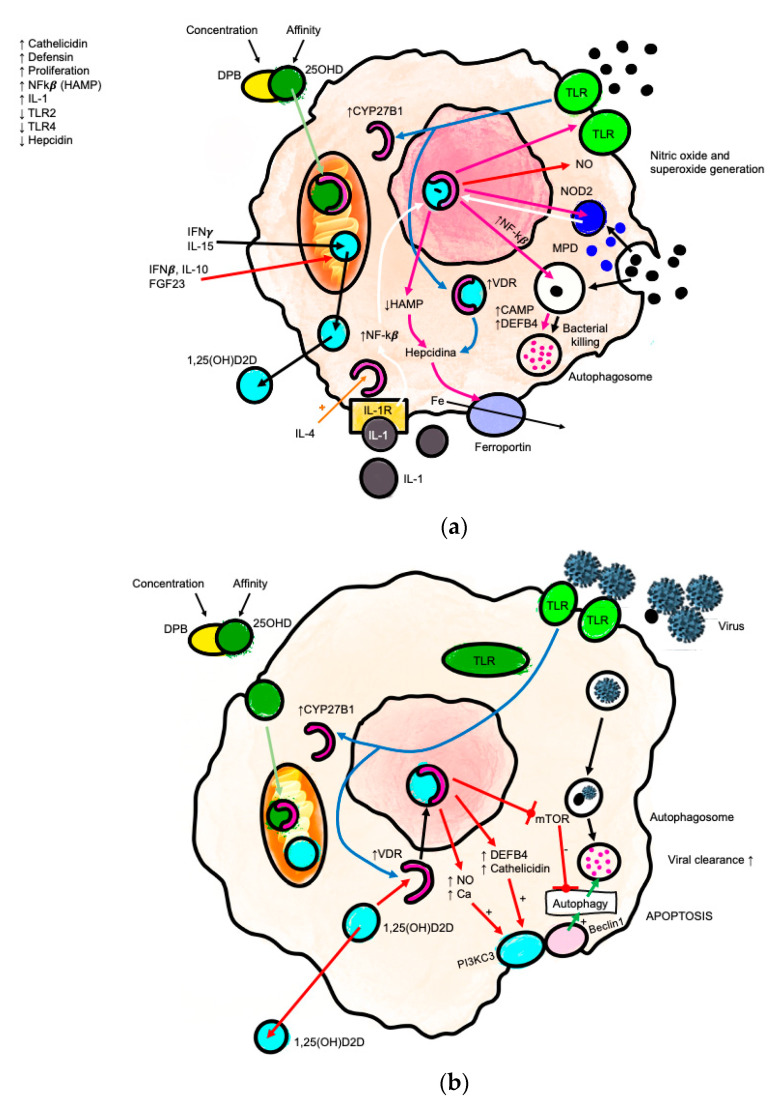
(**a**) Mechanisms for the induction of VDES-mediated antibacterial responses in monocytes. Pattern recognition receptors (TLR2/1) detect germs to induce the expression of CYP27B1 and VDR. The resulting intracrine system for vitamin D (blue arrows) converts calcifediol (25-hydroxyvitamin D (25OHD)) into calcitriol (1,25-dihydroxyvitamin D (1,25(OH)2D)), which then binds to the VDR and promotes transcriptional regulation. Responses to intracrine activation of the VDES (pink arrows) include the following: induction of antibacterial cathelicidin (CAMP) and β-defensin 2 (DEFB4); iron-regulating hepcidin (HAMP) suppression; promotion of autophagy; induction of NOD2 expression; feedback regulation of toll-like receptor expression (TLR); and increased destruction of bacteria. For some responses, accessory immune signals (MDP-noD2 binding and IL-1 responsiveness) cooperate with the intracrine VDES through nuclear factor κB (NF-κB) (white arrows). (**b**) Antiviral actions of the VDES and the innate immune response: autophagy/apoptosis. Autophagy is a fundamental biological process that maintains cellular homeostasis through the encapsulation of the intracellular membrane of damaged organelles and misfolded proteins. Autophagy is also an essential mechanism by which cells cope with viruses. Autophagic encapsulation of viral particles packages them for lysosomal degradation and subsequent presentation of antigens and adaptive antiviral immune responses. Therefore, autophagy may be highly sensitive to changes in serum 25(OH)D levels. The specific mechanisms that enable the VDES to promote autophagy involve downregulation of the mTOR pathway, which inhibits autophagy, and promotion of Beclin 1 and PI3KC3, key autophagy-driving enzymes. The upregulation of intracellular Ca and NO by VDES also stimulates the activity of PI3KC3 to promote autophagy.

**Figure 4 nutrients-14-02447-f004:**
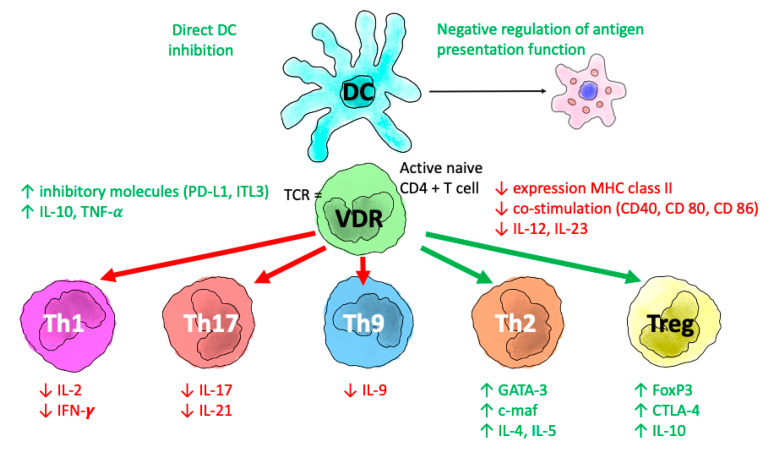
Immunomodulatory actions of calcitriol (1,25 dihydroxy vitamin D3). 1-25OH-D3 exerts effects through the VDR on T lymphocytes and antigen-presenting cells (APCs, in this case the dendritic cell, DC). The effect is upregulation of direct inhibition of APCs and downregulation of antigen presentation. On T lymphocytes, the direct effect consists of induction of T helper-2 lymphocytes (Th2) and regulatory T lymphocytes (Tregs; represented in green text), together with downregulation of T helper-1 (Th1), T helper-17 (Th17), and T helper-9 (Th9) lymphocytes (represented in red text). Abbreviations: IL: interleukin; IFN-γ: interferon-γ; TNF-α: tumor necrosis factor-α; ILT-3: immunoglobulin-3-like transcription; GATA-3: GATA-3 binding protein; FoxP3: forkhead box P3, CTLA-4: protein 4 associated with cytotoxic T lymphocytes.

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
