# Peer review of "Calcifediol for Use in Treatment of Respiratory Disease"

_nutrients, 2022, doi:10.3390/nu14122447_

Round 1

Reviewer 1 Report

Manuscript ID: nutrients-1685058
Type of manuscript: Review
Title: Calcifediol for use in treatment of respiratory disease

This is a an interesting paper considering the role of endocrine system of vitamin D with special regard to calcifediol in treatment of various respiratory diseases mainly asthma, COPD and Covid 19.

Overall the article is good, however I have several remarks that need to be addressed to improve the article.

  1. In Figure 1, the spelling of many words needs to be corrected, for example: 7-dehidrocolesterol, colecalciferol, dihidroxi-, hidroxilasa. This is not English spelling, please correct.
  2. Line 53-54; “ The vitamin D3 produced in the skin is transported directly through the blood 53 bound mainly to vitamin D-binding protein (DBP) and is encoded by the GC gene 54 (CYP2R1/CYP27A1).” What is encoded by GC gene 54 (CYP2R1/CYP27A1?

I think the enzyme 25-hydroxylase, not vit. D3 or DBP. Please correct.

  1. Section 2. Lines 139-157. This is pretty vague information about the effects of vit. D on Treg cells or IgE antibodies. I would expect more detailed information on this topic including mechanisms and with adequate citations.
  2. Line 281; “ the risk” repeats two times.
  3. Conclusions are missing.
  4. 3a, some text information in Figure are in Spanish.
  5. Figure 4. Very poor figure sharpness.
  6. 484-485, “effects deleterious effects”. Remove one of the effects.
  7. Lines 542-552. I'm having trouble understanding this passage and what the authors were trying to convey: “In patients with serum 25(OH)D levels >30 ng/mL who received cholecalciferol, SARS-CoV-2 infection was more frequent (HR 0.66; 95%CI 0.57–0.77), and the risk of severe Covid-19 was higher (HR 0.72; 95% CI 0.52–1.00), as was Covid-19 mortality (HR 547 0.66, 95% CI 0.46–0.93)”. More frequent and higher compared to what? Because in next sentence you wrote that: “These parameters were significantly lower than in patients with  vitamin D deficiency (25(OH)D.” Please rewrite it.
  8. I also have the impression that it would be useful to have this text read and corrected by a native English speaker.

Author Response

1. In Figure 1, the spelling of many words needs to be corrected, for example: 7-dehidrocolesterol, colecalciferol, dihidroxi-, hidroxilasa. This is not English spelling, please correct.

            By mistake, a version of figure 1 with the Spanish spelling was included in the manuscript. We deeply regret the error. We have corrected it and included the correct English spelling in figure 1 of the revised manuscript.

2. Line 53-54; “The vitamin D3 produced in the skin is transported directly through the blood 53 bound mainly to vitamin D-binding protein (DBP) and is encoded by the GC gene 54 (CYP2R1/CYP27A1).” What is encoded by GC gene 54 (CYP2R1/CYP27A1?I think the enzyme 25-hydroxylase, not vit. D3 or DBP. Please correct.

            This is a transcription error. We thank you for the indication and delete (CYP2R1/CYP27A1) in the revised manuscript.

3.  Section 2. Lines 139-157. This is pretty vague information about the effects of vit. D on Treg cells or IgE antibodies. I would expect more detailed information on this topic including mechanisms and with adequate citations.

            As indicated by the reviewer, the authors provide more detailed information supported by appropriate references. Information on the effect of vitamin D on Treg cells and on IgE production has been expanded. The necessary references have been added.

4. Line 281; “the risk” repeats two times.

      One of the two "the risk" has been removed in the revised Ms.

5. Conclusions are missing.

            In the template included in the presentation rules, it is specified that the conclusions section "is not mandatory but can be added to the manuscript if the discussion is unusually long or complex." Our article is a review and, as such, its structure differs from that of an original, so it was not initially included. However, if the reviewers and editor consider it essential, we will include it.

6. 3a, some text information in Figure is in Spanish.

            We deeply regret the error. We have corrected it and included the correct English spelling in figure 1 of the revised manuscript.

7. Figure 4. Very poor figure sharpness.

     The figure is now with better resolution.

8. 484-485, “effects deleterious effects”. Remove one of the effects.

            One of the two "effects" has been removed in the revised Ms.

9. Lines 542-552. I'm having trouble understanding this passage and what the authors were trying to convey: “In patients with serum 25(OH)D levels >30 ng/mL who received cholecalciferol, SARS-CoV-2 infection was more frequent (HR 0.66; 95%CI 0.57–0.77), and the risk of severe Covid-19 was higher (HR 0.72; 95% CI 0.52–1.00), as was Covid-19 mortality (HR 547 0.66, 95% CI 0.46–0.93)”. More frequent and higher compared to what? Because in next sentence you wrote that: “These parameters were significantly lower than in patients with vitamin D deficiency (25(OH)D.” Please rewrite it.

            The paragraph has been modified and its location in the manuscript has been modified (lines 1000 and following) to make it more understandable.

            New text:

            “Cholecalciferol supplementation was associated with slight protection from SARS-CoV2 infection (HR 0.95 [CI 95% 0.91-0.98], p = 0.004). Patients on cholecalciferol treatment achieving 25OHD levels ≥ 30 ng/ml had lower risk of SARS-CoV2 infection, lower risk of severe COVID-19 and lower COVID-19 mortality than non-supplemented 25(OH)D-deficient patients (HR 0.66 [CI 95% 0.46-0.93], p = 0.018). Calcifediol use was not associated with reduced risk of SARS-CoV2 infection or mortality in the whole cohort. However, patients on calcifediol treatment achieving serum 25(OH)D levels ≥ 30 ng/ml also had lower risk of SARS-CoV2 infection, lower risk of severe COVID-19, and lower COVID-19 mortality compared to 25(OH)D-deficient patients not receiving vitamin D supplements (HR 0.56 [CI 95% 0.42-0.76], p < 0.001)”.

10. I also have the impression that it would be useful to have this text read and corrected by a native English speaker.

            The authors have received editorial assistance from Content Ed Net in the preparation of this manuscript and this is noted in the acknowledgements section.

Reviewer 2 Report

The present narrative review summarizes data regarding vitamin D, lung function, and pulmonary diseases such as asthma, COPD, and COVID-19. Moreover, it is hypothesized that calcifediol (25OHD) may be a good option for raising 25OHD quickly. Altogether, it is concluded that 25OHD status may have beneficial effects on the clinical course of respiratory diseases, including COVID-19.

There are several issues that have to be addressed:
Introduction
•    Although a description of vitamin D metabolism and function in lung tissue is meaningful, large parts of the description of vitamin D metabolism, e.g. in the Introduction section, belong to textbook knowledge and should be shortened substantially. 

•    Lines 90-91: This reviewer agrees with the statement that there is no clear consensus on the optimal serum level of 25(OH)D. However, this reviewer disagrees with their statement that a value of ≥30 ng/mL has been considered necessary to ensure optimal health. Note that several nutrition societies such as IOM, NORDEN, and European Food Safety Authority consider a target 25OHD concentration of 50 nmol/l as adequate.

•    Lines 38-39: The reviewer disagrees with the statement that vitamin D is not a vitamin. Rather, it is a hybrid of a vitamin and a prohormone. Note that many groups of people rely on oral intake of vitamin D, such as newborns, people living within the Arctic Circle, and nursing home residents. Moreover, for dark-skinned immigrants in northern countries, veiled women, and people with low outdoor activities, the vitamin character of vitamin D also comes to the fore. By the way, it is surprising that a manuscript on vitamin D is submitted to a nutrition journal questioning the vitamin character of the substance the manuscript is about. 

•    Lines 54-55: It is incorrect that the GC gene is identical with CYP2R1/CYP27A1. 

•    The English of the abstract and of parts of the Introduction is good for reviewing, but not for publication. The manuscript should be checked with a native English speaker.

Paragraphs Asthma and COPD
•    The paragraphs on asthma and COPD can be shortened substantially, especially the part dealing with earlier study results. In fact, they should refer to the results of a recent rigorous umbrella review by Maretzke et al. (Nutrients 2020) on several selected extraskeletal diseases, including asthma and COPD.  

•    Beneficial vitamin D effects by treating asthma have primarily been seen in children. This should be mentioned in the manuscript.  

•    Regarding COPD, they mention that the results observed for cholecalciferol could be extrapolated to calcifediol. However, they should be cautious with such a statement. The literature is replete of scientific hypotheses which later on could not be confirmed by clinical trials. 
COVID-19
•    Lines 472-474: combine to 86-118, and 119-131.

•    Line 504-505: This reviewer disagrees with the statement that three studies shows that treatment with calcifediol could reduce the risk of morbidity and mortality in patients with Covid. Two studies were cohort studies which are not able to prove causality. One of these two studies compared vitamin D vs. 25(OH)D with no control group. In the small RCT they mentioned, important baseline characteristics differed significantly or tended to differ. This can substantially have influenced outcome results of the RCT. Most importantly, however, they failed to refer to a large RCT on vitamin D and COVID-19 not reporting beneficial effects of vitamin D administration (Murai et al. JAMA 2021). 

•    Altogether, their conclusions regarding vitamin D and COVID-19, and regarding beneficial effects of calcifediol are too speculative.    
Other remarks
•    Two important aspects for a reader of a nutrition journal such as “Nutrients” are completely missing: First, to what extent do oral vitamin D and 25OHD doses differ for achieving a given circulating 25OHD concentration. Second, the legal situation in different countries regarding the use of 25OHD for improving vitamin D status would be interesting. Note that in many countries over-the-counter (OTC) vitamin D supplements are available, whereas OTC 25OHD supplements are not. Can calcifediol really be an alternative to vitamin D supplements?

Author Response

Dear Rewiever,

The authors would like to thank the reviewers for their comments, suggestions, and constructive criticism, which will undoubtedly contribute to improving the quality of the manuscript.

Thank you very much.

There are several issues that have to be addressed:

  1. Introduction
    Although a description of vitamin D metabolism and function in lung tissue is meaningful, large parts of the description of vitamin D metabolism, e.g. in the Introduction section, belong to textbook knowledge and should be shortened substantially. 

            This manuscript, in addition to researchers and regular readers of Nutrients, is intended to reach pneumologists and general practitioners interested in this topic. So, after careful review of the pros and cons we have tried to make the knowledge of the endocrine system metabolism of vitamin D more accessible to readers less familiar with the subject. Therefore, we ask the reviewer and editors for permission to keep the introduction longer than usual.

  1. Lines 90-91: This reviewer agrees with the statement that there is no clear consensus on the optimal serum level of 25(OH)D. However, this reviewer disagrees with their statement that a value of ≥30 ng/mL has been considered necessary to ensure optimal health. Note that several nutrition societies such as IOM, NORDEN, and European Food Safety Authority consider a target 25OHD concentration of 50 nmol/l as

We fully agree. A very precise indication from the reviewer. In the new manuscript, the text is modified to make this concept more precise, following the reviewer's indications.

The text “Although calcitriol is the active hormone, it is widely accepted that the serum level of 25(OH)D, or calcifediol, is the biomarker of nutritional VDES status. Although there is no clear consensus on the optimal serum level of 25(OH)D, a value of ≥30 ng/mL (75 nmol/L) has been considered necessary to ensure optimal health [11].

In the new manuscript it is replaced by: Although calcitriol is the active hormone, it is widely accepted that the serum level of 25(OH)D, or calcifediol, is the biomarker of nutritional VDES status[11]. Nevertheless, there is no clear consensus on the optimal serum level of 25(OH)D. The US Institute of Medicine (IOM) (subsequently renamed the National Academy of Medicine), in 2011 specify a serum 25(OH)D level of ≥50 nmol/l (20 ng/mL) as the lower target value for an adequate vitamin D supply, shortly after the D-A-CH nutrition societies (D-A-CH: Germany, Austria, Switzerland), the Scandinavian nutrition societies, the German Osteology governing body (DVO) and the European Society for Clinic and Economic Aspects of Osteoporosis and Osteoarthritis) adopted the IOM guidelines. In contrast, the Endocrine Society and the International Osteoporosis Foundation consider an adequate vitamin D supply to be guaranteed at levels of at least 75 nmol/l (30 ng/ml) Several medical societies and non-governmental organizations such as the American Geriatrics Society (AGS) and the International Osteoporosis Foundation (IOF) have adopted the ES guidelines.

  1. Lines 38-39: The reviewer disagrees with the statement that vitamin D is not a vitamin. Rather, it is a hybrid of a vitamin and a prohormone. Note that many groups of people rely on oral intake of vitamin D, such as newborns, people living within the Arctic Circle, and nursing home residents. Moreover, for dark-skinned immigrants in northern countries, veiled women, and people with low outdoor activities, the vitamin character of vitamin D also comes to the fore. By the way, it is surprising that a manuscript on vitamin D is submitted to a nutrition journal questioning the vitamin character of the substance the manuscript is about. 

The authors conceptually share the reviewer's view, that in extreme situations in which vitamin D cannot be synthesized in the amount necessary to fulfil its physiological functions, it meets all the criteria to be considered a vitamin. In the new manuscript we express it more appropriately

The statement “Vitamin D is not a vitamin, although it was erroneously named as such when it was discovered just over a century ago. In fact, it is a threshold nutrient….” was changed in the revised manuscript by “Vitamin D is a threshold nutrient, which is conditionally indispensable, and therefore meets the criteria to be considered a vitamin in situations where endogenous synthesis is profoundly impaired (such as in vulnerable groups of people, including newborns, infants, people who cover their skin for religious or cultural reasons or those with dark skin, especially if they live in northern countries and elderly people confined to the home, such as nursing home residents, etc.). In fact, part of an endocrine system…..”

  1. Lines 54-55: It is incorrect that the GC gene is identical with CYP2R1/CYP27A1. 

This is a transcription error. We thank you for the indication and delete (CYP2R1/CYP27A1) in the revised manuscript.

  1. The English of the abstract and of parts of the Introduction is good for reviewing, but not for publication. The manuscript should be checked with a native English speaker.

            The authors have received editorial assistance from Content Ed Net in the preparation of this manuscript and this is noted in the acknowledgements section

  1. Paragraphs Asthma and COPD:
    The paragraphs on asthma and COPD can be shortened substantially, especially the part dealing with earlier study results. In fact, they should refer to the results of a recent rigorous umbrella review by Maretzke et al. (Nutrients 2020) on several selected extraskeletal diseases, including asthma and COPD.  

The study published in Nutrients in 2020 by Maretzke et al has been taken into account for our work. A paragraph has been included in the asthma section where the results are discussed.

  1. Beneficial vitamin D effects by treating asthma have primarily been seen in children. This should be mentioned in the manuscript.  

It has been included in the text that the benefits of vitamin D have been seen mainly in children.

Beneficial vitamin D effects by treating asthma have primarily been seen in children. Maretzke et al concluded in 2020 that adequate serum levels of vitamin D in children could reduce the risk of asthma exacerbations, while in the adult population the data are insufficient to draw reliable conclusions [46].

  1. Regarding COPD, they mention that the results observed for cholecalciferol could be extrapolated to calcifediol. However, they should be cautious with such a The literature is replete of scientific hypotheses which later on could not be confirmed by clinical trials. 

Although the original sentence is written in conditional, a modification has now been introduced, indicating that this has not yet been confirmed in clinical trials.

  1. COVID-19
    Lines 472-474: combine to 86-118, and 119-131.

            Corrected.

  1. Line 504-505: This reviewer disagrees with the statement that three studies shows that treatment with calcifediol could reduce the risk of morbidity and mortality in patients with Covid. Two studies were cohort studies which are not able to prove causality. One of these two studies compared vitamin D vs. 25(OH)D with no control group. In the small RCT they mentioned, important baseline characteristics differed significantly or tended to differ. This can substantially have influenced outcome results of the RCT. Most importantly, however, they failed to refer to a large RCT on vitamin D and COVID-19 not reporting beneficial effects of vitamin D administration (Murai et al. JAMA 2021). 

Our manuscript focused primarily on the use of calcifediol (and not vitamin D or other metabolites in the system). Without going into the characteristics of the studies, a critical point to note versus Marai et al Jama 2021 is the advantage of using calcifediol at the doses used because serum 25OHD concentrations increase within a few hours when calcifediol is administered. In fact, in the studies using calcifediol, the dose of calcifediol used in the first week in the studies cited was 1,064 mg (0.532 in the first two days), providing a high availability of circulating 25OHD3 suitable for use in target organs within hours of administration.

  1. Altogether, their conclusions regarding vitamin D and COVID-19, and regarding beneficial effects of calcifediol are too speculative.    

The observational data reported and described in our manuscript strongly suggest that calcifediol supplementation may decrease the severity of this disease, as evidenced by reduced need for intensive care and decreased risk of mortality.  The authors share that more large, randomised trials, designed free of confounding factors, are needed. In the meantime, however, these data support the rapid correction of 25OHD deficiency in all subjects possibly exposed to this coronavirus with Calcifediol. This cost-effective and widely available treatment could have positive implications for the management of COVID-19 worldwide.

  1. Other remarks
    Two important aspects for a reader of a nutrition journal such as “Nutrients” are completely missing: First, to what extent do oral vitamin D and 25OHD doses differ for achieving a given circulating 25OHD concentration. Second, the legal situation in different countries regarding the use of 25OHD for improving vitamin D status would be interesting. Note that in many countries over-the-counter (OTC) vitamin D supplements are available, whereas OTC 25OHD supplements are not. Can calcifediol really be an alternative to vitamin D supplements?

The submitted manuscript does not aim to compare cholecalciferol (vitamin D3) and calcifediol. However, oral calcifediol causes a more rapid rise in serum 25OHD compared to oral cholecalciferol. Secondly, oral calcifediol is more potent than cholecalciferol, so lower doses are needed. Based on RCT results comparing physiological doses of oral cholecalciferol with oral calcifediol, calcifediol is 3.2 times more potent than oral cholecalciferol. In fact, using doses ≤ 25 μg/day, serum 25OHD increased by 1.5 ± 0.9 nmol/l for every 1 μg of cholecalciferol, whereas for oral calcifediol it was 4.8 ± 1.2 nmol/l. Moreover, oral calcifediol has a higher intestinal absorption rate and this may have important advantages in case of decreased intestinal absorption capacity due to a variety of diseases. Furthermore, oral calcifediol has a linear dose-response curve, regardless of the initial serum 25OHD, whereas the increase in serum 25OHD is smaller after oral cholecalciferol, when the initial serum 25OHD is higher. Finally, intermittent intake of calcifediol results in a fairly stable serum 25OHD compared to greater fluctuations after intermittent oral cholecalciferol. Quesada-Gomez JM, Bouillon R.Is calcifediol better than cholecalciferol for vitamin D supplementation? Osteoporos Int. 2018 Aug;29(8):1697-1711. doi: 10.1007/s00198-018-4520-y

Therefore, in respiratory pathologies such as asthma or chronic obstructive pulmonary disease, calcifediol could be the treatment of choice to adjust 25OHD levels rapidly and safely, obviating hydroxylation at the 25OHD position, etc....

Unfortunately. We could not answer the reviewer's question about the legal regulatory situation of cholecalciferol vs. calcifediol in the various countries; thus, in the European Union as far as we know cholecalciferol as a nutrient can be sold as an over-the-counter (OTC) supplement whereas calcifediol is a prohormone, (the prohormone of the vitamin D endocrine system) and therefore its sale to the public must be carefully regulated, requiring a prescription. But we believe that this aspect is beyond the scope of this manuscript.

Round 2

Reviewer 1 Report

I think the paper has been significantly improved and the authors have taken my comments into account. However, I think the few sentences summarizing the review are needed. 

Author Response

We have included a conclusion in the article to summarize the main ideas of the article as indicated by the reviewer.

Reviewer 2 Report

This reviewer  disagrees with their conclusion in the abstract ‘However, in the meantime, rapid correction of 25(OH)D deficiency is recommended for patients with respiratory disease.‘. First, neither COPD, nor asthma or COVID-19 are primarily vitamin D-deficiency diseases. Disease prevention, but not rapid correction of 25(OH)D deficiency, should be the method of choice. Second, Denlinger et al. (Am J Respir Crit Care Med. 2016;193:634-41), in an RCT, have demonstrated a significantly higher risk of respiratory tract infections in patients with in-study 25(OH)D concentration > 75 nmol/l compared to patients with 25(OH)D < 75 nmol/l. Therefore, in the clinical setting, caution is necessary when administering drugs which have no approval for the treatment of a disease. Efficacy and safety of calcifediol have to be clearly demonstrated by RCTs, including approval for the treatment of patients with pulmonary diseases by national and international drug administrations.   

With a few exceptions, the paragraphs dealing with COPD and asthma obviously remained unchanged, although this reviewer has suggested that the paragraphs should be shortened substantially.

Although this manuscript focusses on calcifediol and not on cholecalciferol, it would have been possible to present in-study 25(OH)D data of the vitamin D RCTs on COPD and asthma, which were included in the umbrella review by Maretze et al.      

Although suggested by this reviewer, they do not refer to the JAMA article by Murai et al., demonstrating no beneficial effect of vitamin D administration on COVID19. In that study, circulating 25(OH)D increased from 51 nmol/l (baseline) to > 100 nmol/l (in-study) in the vitamin D supplemented group, whereas 25(OH)D remained largely constant at 51 nmol/l in the placebo group. Thus, in the vitamin D supplemented group baseline 25(OH)D was in the range the authors of the present study consider inadequate (< 75 nmol/l) and increased in the range they consider adequate (> 75 nmol/l). It is unclear to this reviewer, which conclusions they draw from these results regarding calcifediol administration.

Several studies have compared calcifediol versus cholecalciferol supplementation in apparently healthy individuals and also in different groups of patients. A manuscript dealing with the apparent benefits of calcifediol administration should refer to the results of these studies. This paragraph should also describe differences in intestinal absorption pathways and circulating 25(OH)D increase between cholecalciferol and calcifediol.

Altogether, a much more cautious conclusion would have been necessary.

Author Response

Author's Reply to the Review Report

This reviewer disagrees with their conclusion in the abstract ‘However, in the meantime, rapid correction of 25(OH)D deficiency is recommended for patients with respiratory disease.‘. First, neither COPD, nor asthma or COVID-19 are primarily vitamin D-deficiency diseases. Disease prevention, but not rapid correction of 25(OH)D deficiency, should be the method of choice. Second, Denlinger et al. (Am J Respir Crit Care Med. 2016;193:634-41), in an RCT, have demonstrated a significantly higher risk of respiratory tract infections in patients with in-study 25(OH)D concentration > 75 nmol/l compared to patients with 25(OH)D < 75 nmol/l. Therefore, in the clinical setting, caution is necessary when administering drugs which have no approval for the treatment of a disease. Efficacy and safety of calcifediol have to be clearly demonstrated by RCTs, including approval for the treatment of patients with pulmonary diseases by national and international drug administrations.  

We agree with the reviewer. Neither asthma, nor COPD, nor COVID-19 are diseases caused by a vitamin-D deficiency and the authors think that nothing in the text suggests otherwise. We only indicate that, according to the arguments of the manuscript, when there is a deficit of 25(OH)D, it must be corrected, regardless of whether the patient has a respiratory pathology. It is not being indicated to treat respiratory disease with something not approved, but only the correction of vitamin-D deficiency.

Regarding the reference indicated by the reviewer, there are several references, cited in the text, that maintain different opinions.

Regarding the fact that the efficacy of calcifediol must be demonstrated and approved by international regulators for the treatment of respiratory diseases, we share that idea with the reviewer, as we have indicated above. We only indicate that it is an option to consider in vitamin D deficiency because of its interesting pharmacokinetic properties, which give calcifediol some advantages over vitamin D: 1) calcifediol induces a more rapid rise in circulating 25OHD than oral cholecalciferol; 2) oral calcifediol is more potent than cholecalciferol; 3) oral calcifediol has a higher rate of intestinal absorption, giving it advantages in cases of malabsorption; 4) oral calcifediol has a linear dose-response curve, independent of initial serum 25OHD; and 5) intermittent administration of calcifediol produces a fairly stable serum 25OHD, compared to fluctuations following intermittent oral cholecalciferol.

Quesada-Gomez JM & Bouillon R. Is calcifediol better than cholecalciferol for vitamin D supplementation? Osteoporos Int. 2018 Aug;29(8):1697-1711. Doi: 10.1007/s00198-018-4520-y.

Cesareo, R.; Falchetti, A.; Attanasio, R.; Tabacco, G.; Naciu, A.M.; Palermo, A. Hypovitaminosis D: Is it time to consider the use of calcifediol? Nutrients 2019, 11, 1016, doi:10.3390/nu11051016.

With a few exceptions, the paragraphs dealing with COPD and asthma obviously remained unchanged, although this reviewer has suggested that the paragraphs should be shortened substantially.

The authors share the reviewer's assertion that the sections on asthma and COPD are lengthy. But the literature on these pathologies is abundant, there are controversial aspects, as evidenced by the comments we receive and we believe that we have tried to review in detail this aspect that, although it may be well known by endocrinologists, internists and other specialties, is not so much for pulmonologists who constitute an important nucleus of potential readers of the article.

Although this manuscript focusses on calcifediol and not on cholecalciferol, it would have been possible to present in-study 25(OH)D data of the vitamin D RCTs on COPD and asthma, which were included in the umbrella review by Maretze et al.  

Indeed, the article focuses on calcifediol as suggested by the guest editors when they called for the Special Issue of nutrients: Vitamin D Endocrine System: Calcifediol for Treatment and Prevention of Infection and Disease). It is therefore not our purpose to also review the aspects related to vitamin D3 in each of the pathologies so as not to extend the article excessively, especially the sections on asthma and COPD. In fact, the reviewer has already pointed out to us that the sections are long as they are; considering this aspect would force us to lengthen the text more. Nevertheless, we have included the excellent reference proposed by the reviewer, so that interested readers can look further into some aspects not included in that manuscript.

Maretzke, F.; Bechthold, A.; Egert, S.; Ernst, J.B.; van Lent, D.M.; Pilz, S.; Reichrath, J.; Stangl, G.I.; Stehle, P.; Volkert, D.; et al. Role of Vitamin D in Preventing and Treating Selected Extraskeletal Diseases-An Umbrella Review. Nutrients 2020, 12, doi:10.3390/NU12040969.

Although suggested by this reviewer, they do not refer to the JAMA article by Murai et al., demonstrating no beneficial effect of vitamin D administration on COVID19. In that study, circulating 25(OH)D increased from 51 nmol/l (baseline) to > 100 nmol/l (in-study) in the vitamin D supplemented group, whereas 25(OH)D remained largely constant at 51 nmol/l in the placebo group. Thus, in the vitamin D supplemented group baseline 25(OH)D was in the range the authors of the present study consider inadequate (< 75 nmol/l) and increased in the range they consider adequate (> 75 nmol/l). It is unclear to this reviewer, which conclusions they draw from these results regarding calcifediol administration.

The article of Murai el al. is cited with the reference number 130 in the paragraph where it is indicated that there are works that find some benefit in adequate serum levels of 25OHD (references 90 to 122) and others that do not find any (references 123-135). In total there are 46 bibliographic references

We do not go further into the description of this or other papers because, as previously indicated, the aim of this work was to focus only on calcifediol and not on other metabolites of the endocrine system of vitamin D such as calcitriol or vitamin D3 itself, which have also been used in COVID-19.

Muray et al's multicentre, double-blind, randomised, placebo-controlled clinical trial in Brazilian hospitals included 240 hospitalised patients with COVID-19 who had moderate to severe COVID-19 expression (the majority). Patients were randomised to receive a single oral dose of 200,000 IU vitamin D 3 (n=120) or placebo (n=120).  Of note, ethnicity was not matched between the arms, and diabetes was 41% in the treatment group versus 29% in the control group. Fifteen per cent of the vitamin D3 treatment group required assisted ventilation already at baseline, compared to 12 per cent of the control group. There was no difference in the risk of death, slightly higher in the vitamin D3-treated group (7.6%) than in the untreated (5.1%), nor in the need for mechanical ventilation 7.6% vs. control 14.4% or the risk of ICU admission 16.0% in the treated vs. 21.2% in the untreated 25%. Hospital stay was also not different in both groups (mean 7 days in both groups).

Without discussing the important differences described between the two arms, which by themselves for some authors could explain the lack of significant differences, we would like to highlight an important aspect that perhaps we did not adequately indicate in our responses to the reviewer, nor in the manuscript. In the Muray et al trial, as the reviewer correctly noted, mean serum 25OHD levels increased significantly in the treated to 44.4 ng/mL versus 19.8 ng/mL in the untreated, but it took two weeks to reach these levels, whereas in calcifediol-treated patients as described in our manuscript, these levels were obtained in the hours immediately following oral administration, without obesity, inflammatory processes, malabsorption etc.... .  This aspect is critical in our opinion because it facilitates the synthesis of calcitriol, the active hormone of the Vitamin D Endocrine System that enables responses in innate and adaptive immunity, etc... critical in the treatment of COVID and in an immediate way. This property is very important for treating COVID-19 because the two main effects of vitamin D regarding COVID-19 are reduced viability and replication of SARS-CoV-2 and reduced risk of a cytokine storm.

Moreover, in Muray's work, it appears that many patients were already at an advanced stage in the natural history of the disease where the endocrine-mediated actions of vitamin D would not be effective.

Following the reviewer's proposal, we added one sentence to the manuscript (Lines 757-767 of the revised manuscript): “These results contrast with those described in the multicentre, double-blind, randomised, placebo-controlled clinical trial by Muray et al [130] that included 240 hospitalised patients with COVID-19 who had moderate to severe COVID-19 expression (the majority). Patients were randomised to receive a single oral dose of 200,000 IU vitamin D3 (n = 120) or placebo (n = 120), finding no difference in the risk of death, need for assisted ventilation or even the number of days of hospitalisation between the two groups.  Treatment with calcifediol has an advantage over vitamin D3 supplementation as it increases serum 25(OH)D concentrations more rapidly, in a matter of hours rather than days (see below). This property is very important for treating COVID-19 because the two main effects of vitamin D regarding COVID-19 are reduced viability and replication of SARS-CoV-2 and reduced risk of a cytokine storm”.

Quesada-Gomez JM, Lopez Miranda J, Entrenas-Castillo M, Nogues y Solans X, Mansur J, Bouillon R. Vita-min D endocrine system and COVID-19. Treatment with calcifediol. Nutrients

Several studies have compared calcifediol versus cholecalciferol supplementation in apparently healthy individuals and also in different groups of patients. A manuscript dealing with the apparent benefits of calcifediol administration should refer to the results of these studies. This paragraph should also describe differences in intestinal absorption pathways and circulating 25(OH)D increase between cholecalciferol and calcifediol.

According to the reviewer's indications we now complete the manuscript by including the potential superiority of calcifediol, including a paragraph on the differences in the absorption pathways of both metabolites, as well as that calcifediol does not require hepatic 25-hydroxylation.

We add in the manuscript several paragraphs to that effect

Calcifediol may have some advantages over vitamin D, which gives it a certain superiority for use in vitamin D deficiency replacement therapy in asthma, COPD, COVID-19 and other respiratory diseases: 1) calcifediol induces a more rapid increase in circulating 25OHD than oral cholecalciferol; 2) oral calcifediol is more potent than cholecalciferol; 3) oral calcifediol has a higher rate of intestinal absorption, which confers advantages in cases of malabsorption; 4) oral calcifediol has a linear dose-response curve, independent of initial serum 25OHD; and 5) intermittent administration of calcifediol produces a fairly stable serum 25OHD, compared to fluctuations following intermittent oral cholecalciferol [160]. Therefore, oral calcifediol is more potent than vitamin D3 according to the results of nine RCTs, comparing physiological doses of oral cholecalciferol with oral calcifediol [161], and in clinical trials where the two drugs have been compared head-to-head [162].

In the absorption process, vitamin D3 is incorporated into chylomicrons and enters the lymphatic system. The chylomicrons enter the bloodstream via the superior vena cava. Most of the vitamin D3 is incorporated into body fat. Vitamin D3 that is slowly released from body fat into the circulation is converted in the liver to calcifediol (25OHD3). This is the likely explanation why vitamin D3 administration takes time to reach a steady-state concentration of 25OHD3.  However, calcifediol is more hydrophilic and therefore, after ingestion, is absorbed into the venous portal system, immediately increasing circulating concentrations of 25(OH)D [160], which is available within hours as a substrate for calcitriol synthesis in broncho-alveolar lung cells, immune cells or other potential target tissues. This ease of absorption and availability [163] is of great importance in patients with severe malabsorption of any aetiology [160].

Furthermore, calcifediol does not require hepatic 25-hydroxylation. Functional impairment of CYP2R1 activity has been reported in patients with obesity, malabsorption [164] or inflammatory lung diseases such as COPD or asthma [165][166].

In summary, calcifediol could play an important role in the treatment of vitamin D deficiency in patients with respiratory diseases. Depending on the availability of clinical trials or not, we can recommend its use with a greater or lesser degree of evidence. Clearly in asthma and with somewhat less evidence in COPD. Maintaining adequate 25(OH)D levels may contribute to better control of both pathologies, avoid exacerbations, and especially avoid exacerbations of infectious origin. In Covid-19, observational data consistently suggest that calcifediol supplementation may decrease the severity of this disease, as evidenced by a reduced need for intensive care and a decreased risk of mortality. However, we will have to wait for the results of large, randomized trials clinical trials for this recommendation to have a high level of evidence. In the meantime, the practical recommendation should be to systematically determine 25(OH)D serum levels in all patients with chronic respiratory pathologies and, in those with deficient levels, correction of the 25OHD deficit should be recommended by substitution treatment.

Altogether, a much more cautious conclusion would have been necessary